# Evaluation of the First Three Years of Treatment of Children with Congenital Hypothyroidism Identified through the Alberta Newborn Screening Program

**DOI:** 10.3390/ijns10020035

**Published:** 2024-05-02

**Authors:** Iveta Sosova, Alyssa Archibald, Erik W. Rosolowsky, Sarah Rathwell, Susan Christian, Elizabeth T. Rosolowsky

**Affiliations:** 1Department of Laboratory Medicine and Pathology, Faculty of Medicine & Dentistry, University of Alberta, Edmonton, AB T6G 2B7, Canada; isosova@ualberta.ca; 2Genetics and Genomics, Alberta Precision Laboratories, Edmonton, AB T6G 2B7, Canada; smc12@ualberta.ca; 3Department of Pediatrics, Faculty of Medicine & Dentistry, University of Alberta, Edmonton, AB T6G 1C9, Canada; alyssa.archibald@ahs.ca; 4Department of Physics, Faculty of Science, University of Alberta, Edmonton, AB T6G 2E1, Canada; erosolow@ualberta.ca; 5Women & Children’s Health Research Institute, Faculty of Medicine & Dentistry, University of Alberta, Edmonton, AB T6G 1C9, Canada; srathwel@ualberta.ca

**Keywords:** neonatal screening, congenital hypothyroidism, neurodevelopment, euthyroid, guidelines

## Abstract

The effectiveness of newborn screening (NBS) for congenital hypothyroidism (CH) relies on timely screening, confirmation of diagnosis, and initiation and ongoing monitoring of treatment. The objective of this study was to ascertain the extent to which infants with CH have received timely and appropriate management within the first 3 years of life, following diagnosis through NBS in Alberta, Canada. Deidentified laboratory data were extracted between 1 April 2014 and 31 March 2019 from Alberta Health administrative databases for infants born in this time frame. Time to lab collection was anchored from date of birth. Timeliness was assessed as the frequency of monitoring of Thyroid Stimulating Hormone (TSH) and appropriateness as the frequency of children maintaining biochemical euthyroidism. Among 160 term infants, 95% had confirmation of diagnosis by 16 days of age. The cohort had a median of 2 (range 0–5) TSH measurements performed in the time interval from 0 to 1 month, 4 (0–12) from 1 to 6 months, 2 (0–10) from 6 to 12 months, and 7 (0–21) from 12 to 36 months. Approximately half were still biochemically hypothyroid (TSH > 7 mU/L) at 1 month of age. After becoming euthyroid, at least some period of hypo- (60%) or hyperthyroidism (TSH < 0.2 mU/L) (39%) was experienced. More work needs to be performed to discern factors contributing to prolonged periods of hypothyroidism or infrequent lab monitoring.

## 1. Introduction

Thyroid hormone is essential to the developing brain and nervous system, especially before 3 years of age. Congenital hypothyroidism (CH) is a condition that describes an infant who is born without the ability to make sufficient thyroid hormone, most commonly due to a structurally abnormal gland or one that cannot make thyroid hormone. The diagnosis of primary CH is made by demonstrating elevated Thyroid Stimulating Hormone (TSH) and reduced free thyroxine (FT4) in serum.

Infants with CH may not display any obvious clinical symptoms during the neonatal period, making it difficult to diagnose. Delays in treatment may result in permanent intellectual disability. Early detection through newborn screening (NBS) and timely initiation of levothyroxine therapy successfully prevent intellectual disability and other negative consequences of untreated hypothyroidism in the affected neonates. The effectiveness of newborn screening for CH in reducing the prevalence of intellectual disability has been incontrovertibly demonstrated [1,2].

In Canada, Quebec was the first province to provide universal newborn screening for CH in April 1974 [3]. In Alberta, this became available in 1977 [4]. The incidence of CH in Alberta has been reported to be approximately 1 in 3500, with 10 to 15 cases detected each year [5].

To be effective at preventing intellectual disability, positive screening results must be followed up appropriately and in a timely manner. A positive newborn screen for CH is confirmed with measurements of serum TSH and FT4 levels. Recommendations from American and European bodies suggest that levothyroxine therapy should be initiated as early as possible and by 14–15 days of age [6,7,8,9]. The goal of therapy is to reach euthyroidism with a normalized FT4 within two weeks and a normal TSH within one month of initiating treatment [6]. There is currently no formal Canadian guideline. Afterwards, the child generally remains on thyroid hormone replacement for at least 2 years and regularly undergoes blood testing to monitor the adequacy of treatment. Persistent hypothyroidism may lead to intellectual disability, while hyperthyroidism can result in adverse effects such as poor weight gain and premature craniosynostosis [10,11,12].

A previous study of the Alberta Newborn Screening Program (ANSP) for CH revealed heterogeneity of practice in diagnosis and proximal management, subsequent to notification of a positive CH screen [13], but there have been no studies detailing the outcomes of long-term monitoring over the subsequent 3 years. The objective of this study was to evaluate the extent to which infants with CH have received timely and appropriate management within the first 3 years of life subsequent to their identification through the ANSP.

## 2. Materials and Methods

Study design: This is a retrospective chart review of Alberta infants who screened positive for and were diagnosed with CH through the ANSP. Deidentified laboratory data were extracted from Alberta Health administrative databases for infants born between 1 April 2014 and 31 March 2019: Ambulatory Care, Inpatient, Laboratory, Pharmaceutical Information, and Population Registry. All outputs were limited to 0 to 3 years of age. For example, if the service event occurred on or after the recipient’s 3rd-year birthday, the record was not included during the data extraction. All confirmed cases of CH identified for infants born during this time frame were included in the initial analyses of incidence and time to first confirmatory TSH measurement.

NBS for CH in Alberta: The ANSP is a province-wide screening program offered to all infants born in Alberta and is located at the University of Alberta Hospital in the provincial capital, Edmonton. The ANSP screens for 22 conditions, including congenital hypothyroidism due to primary causes. The NBS for CH is performed by measuring TSH levels from capillary dried blood specimens (DBSs) collected on Whatman 903 filter paper. Collection of the DBS is recommended at 24 to 72 h of age, as previously described [4]. During the period of this study, the TSH eluted from a single 3.2 mm DBS was analyzed through AutoDELFIA^®^ Neonatal hTSH time-resolved fluoroimmunoassay (PerkinElmer, Turku, Finland), using AutoDELFIA instruments with the corresponding software (Multicalc 2.1 Rev3, PerkinElmer, Turku, Finland), in full compliance with the manufacturer’s instructions.

The CH screening algorithm has two types of out-of-range results: borderline or critical. The borderline cutoff is age-related (0–30 h, TSH ≥ 32 mU/L; 31–191 h, TSH ≥ 26 mU/L; and ≥192 h, TSH ≥ 20 mU/L). The critical cutoff (TSH ≥ 50 mU/L) is the same for all age groups. Infants with borderline results on the initial specimen are followed up with a second NBS collection, whereas infants with critical results are immediately referred to a pediatric endocrinology specialist for a confirmatory diagnostic evaluation. Two borderline results (on separate specimens) are followed up as a critical result. Since CH may be missed in infants with low birth weight (BW) due to delayed TSH rise if screened early after birth, all infants with BW < 2000 g are recommended to have blood spots recollected at 21–28 days of age, even if the initial screen result was normal [4].

Timeliness of diagnosis: Timeliness of diagnosis was described as the time from birth to the date of confirmatory laboratory investigation of serum TSH and FT4. The most sensitive test for detecting primary CH is measurement of TSH [14]. Therefore, we focused our analyses on frequency and levels of TSH measurements.

Timeliness of monitoring: Timeliness of monitoring was assessed as the frequency of monitoring of thyroid function indices. It was anchored to the date of birth of the infant, rather than to the date of the abnormal NBS, because recommendations for treatment initiation and monitoring are expressed in terms of the infant’s age. In 2013, the ANSP (then known as the Alberta Newborn Metabolic Screening Program) developed the “Clinical Algorithm for CH Abnormal Screen Result”, which was based on recommendations from the American Academy of Pediatrics, the Lawson Wilkins Pediatric Endocrine Society (now known as the Pediatric Endocrine Society), and the American Thyroid Association [6,15,16]. TSH +/− FT4 levels should be monitored as follows:

At 2 weeks and then 4 weeks from 0 to 1 month old (≥2 times);

Every 1 to 2 months for infants 1 to 6 months old (≥3 times);

Every 3 to 4 months for infants 6 to 36 months (≥7 times).

Appropriateness of treatment: The goals are to normalize serum FT4 levels within 2 weeks of age and serum TSH levels within 1 month of age. Once FT4 levels are normal, TSH should be used to adjust doses, aiming for TSH levels in the lower half of the normal range. The Alberta Clinical Algorithm recommends levels between 0.5 and 2 mU/L by one month of age and thereafter [16]. Infants born between 1 January and 31 March 2019 were not included in this portion of the analysis because there were insufficient data.

On 12 April 2016, the reference intervals for TSH and FT4 were aligned throughout Alberta according to the performing methodology [17]. Because of the immediate postnatal TSH surge, TSH and thyroid hormones are higher in infants and subsequently fall, approximating adult levels by about 1 month of age [18]. Before this alignment, some regional labs used an upper-limit TSH of 6.8 for infants 8 days to 1 year and 6.5 mU/L for age ≥ 1 year. The lower limit of the normal range of TSH was 0.2 mU/L. For our analyses, we defined biochemical hypothyroidism as TSH > 7 and hyperthyroidism as TSH < 0.2 mU/L. Subclinical hypothyroidism or hyperthyroidism occurs when the TSH level is increased or decreased, respectively, above the normal range, but the FT4 level is normal.

Premature infants can have a blunted or delayed TSH surge due to immaturity of their hypothalamic-pituitary-thyroidal axis [19]. In all premature infants with BW < 2000 g, their NBS are repeated between 21 and 28 days of age if the initial screen was normal. We include all confirmed cases of CH, including premature infants, when reporting the incidence of confirmed CH during the specified time frame. Subsequently, we excluded premature infants from further analyses because their timing and frequency of monitoring would skew the overall time-based results. (In addition, these infants were cared for in Neonatal Intensive Care Units, which is not reflective of most infants with CH who are monitored in the outpatient, community setting).

Statistical analysis: All statistical analyses were performed using SAS Ver. 9.4 and R Ver. 4.1.1. We computed the mode and the median because we were interested in both the most common values and the middle values. Categorical variables are presented as frequencies and percentages. Stratification variables include year of birth and infant age (in days) where appropriate.

## 3. Results

### 3.1. Timeliness of Diagnosis

Between 1 April 2014 and 31 March 2019, 273,319 infants were registered in Alberta, and out of these, 271,826 received NBS. Out-of-range (i.e., positive) NBS results for CH were reported for 247 babies. Further diagnostic testing revealed 185 true-positive NBS results for CH, an incidence of 1:1470.

Infants most commonly were 6 or 7 days of age at the time of the first confirmatory TSH measurement. Most infants (84%) had biochemical confirmation of diagnosis within 16 days of age (range 1 to 69 days). Among cases where borderline NBS results required a second or third repeat screening, confirmation of hypothyroidism by venous sampling typically occurred within one week from when the overtly positive NBS was reported.

When premature infants or infants who died were removed from the analysis, there were 160 infants who had confirmation of positive screens by 2 to 45 days, with 95% of infants being diagnosed within 16 days of age (Table 1 and Figure 1). In 4 cases, a venous FT4 was collected (not TSH) before day 6, and the results were sufficiently low to prompt treatment. (TSH levels were done afterwards and remained high, verifying CH due to primary causes.) In addition, there was one infant with several equivocal NBS with first diagnostic TSH at day 28, one infant with a positive NBS confirmed one month later on day 45, and two infants who did not have diagnostic testing until days 17 and 19 for unknown reasons.

### 3.2. Timeliness of Monitoring

The effectiveness of longitudinal treatment was determined through serum TSH measurements for those with eligible lab tests within the pre-defined study time frame (Figure 2). The median (mode; range) number of TSH measurements during these times periods was 2 (2; 0 to 5) from 0 to 1 month of age, 4 (5; 0 to 12) from 1 to 6 months of age, 2 (2; 0 to 10) from 6 to 12 months of age, and 7 (5; 0 to 21) from 12 to 36 months of age.

Within each year, there was at least one infant with infrequent laboratory monitoring. In 2014, one infant with an initial venous TSH of >150 mU/L had no subsequent labs between days 8 and 63 of age. In 2015, an infant with an initial TSH of 523 mU/L had no additional labs between days 6 and 315. In 2016, an infant with an initial TSH of 984 mU/L at 6 days of age had no subsequent labs. In 2017, there was an infant who did not have any labs past 17 days of age, and the TSH was 88 mU/L at that time. In 2018, one infant had no labs beyond 43 days of age despite being hypothyroid (TSH was 18 mU/L).

### 3.3. Appropriateness of Treatment

The frequencies of euthyroidism by the end of the first month of age are depicted in Figure 3 and Figure 4. Overall, 42% had biochemical euthyroidism, but up to 51% of infants still manifested biochemical hypothyroidism at one month of age. Almost all of these were cases of subclinical hypothyroidism. The highest TSH levels at one month of age were observed among infants with the highest initial pretreatment TSH levels. Several among them still had TSH levels of ≥100 mU/L at one month of age. Five percent of the cohort was biochemically hyperthyroid at the end of the first month of age.

Once euthyroidism had been achieved, up to 67% of infants experienced some period of hypothyroidism (Table 2). Almost all of them had mild TSH elevations (with normal free T4 levels); they were followed up with timely monitoring and subsequently demonstrated to be within normal range. However, there were exceptions among this cohort who experienced protracted periods of severe hypothyroidism:
In 2014, there was one infant who experienced prolonged hypothyroidism. This infant’s initial venous TSH was >500. This infant’s TSH level did not fall below 10 mU/L until day 437. Thereafter, this infant became euthyroid, which persisted until the last record at 1041 days.One infant born in 2015 did not become euthyroid until 89 days of age. Afterwards, this infant had fluctuating levels of hypothyroidism and euthyroidism on day 120 (TSH 24), became euthyroid on days 616 (TSH 126) to 734 (TSH 129), and then was euthyroid on days 778 (TSH 24) to 810 (TSH 23 mU/L).In 2016, an infant with a free T4 of 6.9 pmol/L was started on levothyroxine for CH. At 360 days of age, the TSH level was >150; and at 505 days of age, the TSH was 91 mU/L. There were no other thyroid function indices in between.

After becoming euthyroid, hyperthyroidism was frequently encountered. All of these were cases of subclinical hyperthyroidism.

**Table 2 IJNS-10-00035-t002:** Hypothyroidism (TSH > 7.0 mU/L) and hyperthyroidism (TSH < 0.2 mU/L) beyond 1 month of age after becoming euthyroid.

Year	2014 *n* = 27	2015 *n* = 35	2016 *n* = 30	2017 *n* = 39	2018 *n* = 23
Hypothyroidism after euthyroidism (*n*; %)	17 (63%)	21 (60%)	20 (67%)	24 (62%)	11 (48%)
TSH range (mU/L)	8.4 to 59.1	7.1 to 129.4	7.6 to >150	8.2 to 191.4	7.7 to 78.2
Hyperthyroidism after euthyroidism (*n*; %)	13 (45%)	22 (56%)	13 (37%)	16 (36%)	6 (21%)

Year 2019 (infants born between 1 January and 31 March 2019) was not included in this analysis because there were insufficient data.

## 4. Discussion

The most recent “Technical Report on the Screening and Management of Congenital Hypothyroidism” from the American Academy of Pediatrics (AAP) stresses the importance of clinical and laboratory follow-up of children with CH [19]. In addition to NBS, the management of CH requires timely confirmation of diagnosis and initiation of treatment and consistent follow-up. The major factors that determine long-term cognitive outcomes include the severity of CH, starting dose and timing of initiation of treatment with levothyroxine, and time to restoration of euthyroidism [20]. The goal is to initiate levothyroxine therapy as early as possible, generally by 2 weeks of age [19]. TSH levels should be kept within the age-specific reference range. There is no head-to-head comparison of one monitoring protocol over another, but normal cognitive outcomes have been achieved with monthly, bimonthly, and 3-monthly measurements in the first 2 to 3 years of life. In addition, it is recommended that more frequent monitoring happen during more rapid periods of growth, for example, in the first year of life [14].

We noticed an increased incidence of CH in Alberta over the last 20 years. Between 2002 and 2005, the incidence of CH in Alberta was reported to be approximately 1 in 3500; however, our study shows an incidence of 1:1470 [5]. An increased global incidence of CH over the past 2–3 decades has been reported also in the province of Quebec in Canada, as well as in France, New Zealand, Italy, Argentina, Greece, and the United States [21,22]. It is not completely clear what is causing this increase; most of these increases have been ascribed to the lowering of TSH cutoffs, therefore allowing for the detection of milder cases of CH. In some studies, changes in the ethnic composition of the screened population were also considered to be the cause of the increase [21,22].

Our study demonstrated that nearly all 160 term infants with CH detected through NBS had confirmation of their diagnosis within 16 days of age. There was an overall decrease in TSH levels between the time of the initial confirmatory serum sample and one month of age. Though we cannot with certainty state that this was due to levothyroxine therapy, the trend implies that treatment had been started. TSH measurements were most commonly measured 2 times between 0 and 1 month of age, 4 times between 1 and 6 months, 2 times between 6 and 12 months, and 7 times from 12 up to 36 months of age. The median counts within each time interval show that TSH monitoring generally happened as recommended by international guidelines.

However, our study also showed that the frequency of laboratory monitoring, as well as the attainment and sustainment of euthyroidism, varied widely. Some infants had a paucity of laboratory monitoring completed, while others experienced venipuncture frequently. We discovered several infants each year who were over 1 month of age before a TSH level was measured, had no laboratory testing beyond the first measurement, or went a year or more without any monitoring. Using retrospective administrative data limits us from accessing the clinical context within which care was happening. For instance, we are unable to ascertain whether infants without subsequent monitoring had moved out of province. Nevertheless, further investigations are required to understand the circumstances of these cases.

Among those who had FT4 data available, almost all of their levels were within normal range by one month of age; however, the TSH levels of nearly half of the cohort exceeded 7 mU/L. In other words, subclinical hypothyroidism was common at 1 month of age and frequently recurred among those who eventually became euthyroid. TSH values above 5 mU/L are considered to be abnormal in infants over 3 months of age [19]. Because the hypothalamic-pituitary-thyroidal axis adapts finely to changes in serum free thyroid hormone levels, an increase in TSH suggests insufficiency of thyroid hormone. The 2020–2021 European guidelines recommend treatment with levothyroxine if the serum TSH concentration is 6 to 20 mU/L (with normal free T4) beyond the age of 21 days [14]. Whether mild subclinical hypothyroidism affects neurocognitive outcomes remains a question requiring further investigation. Gross and Van Vliet found that infants with recurrent episodes of subclinical hypothyroidism after 6 months of age had cognitive scores lower than their peers, a higher incidence of behavioral issues, and poorer school performance [2].

Higher initial doses of levothyroxine increase FT4 levels more rapidly but can lead to transient periods of hyperthyroidism. A retrospective study of a Polish CH newborn screening program between 2017 and 2021 demonstrated that 51 (58%) out of 88 patients were optimally treated, while approximately 25% were overtreated. The authors of this study suggested that overtreatment may have occurred among those with transient or milder forms of hypothyroidism who did not require as much exogenous thyroid hormone [23]. A belief exists among some clinicians that overtreatment may be more acceptable than undertreatment, but overtreatment may be associated with inattention, hyperactivity, anxiety, and deficits in cognitive function [24]. Long-term behavioral problems have been described in some cases, but the data are inconsistent with regards to the types and severity of behavioral problems.

The retrospective nature of our study, using data from administrative databases, precludes us from explaining the causes of infrequent TSH monitoring for some infants or why nearly half of the cohort experienced subclinical hypothyroidism. It can be inferred that the overall lowering of TSH levels after confirmation of diagnosis reflects that therapy was started; however, the timing of when levothyroxine therapy was prescribed and started could not be accurately ascertained from the administrative databases and is a limitation of this study. While inadequate therapy could be one reason for suboptimal outcomes, the reasons are likely to be multifactorial. These include the etiology of congenital hypothyroidism, with thyroid agenesis, ectopia, or hypoplasia likely to result in more severe disease; limited access to healthcare providers, laboratory facilities, and treatments; parental non-adherence due to a misunderstanding about the importance of treatment; and providers’ lack of knowledge in management goals or confidence in interpreting lab results. Based on health insurance claims databases in the United States, Kemper and colleagues identified that approximately 40% of children with CH appeared to have discontinued treatment by 36 months after initiation of treatment [25]. Up to half of patients with a diagnosis of CH may be lost to follow-up [25,26,27]. The Region 4 Midwest Genetics Collaborative established the Congenital Hypothyroidism 3-Year Follow-Up Workgroup in 2011. From the Clinician Survey, clinicians reported that some patients had mild or transient hypothyroidism and had normal thyroid testing off medication; however, some parents had stopped treatment on their own. From the Parent Survey, only two-thirds of parents replied that they were satisfied with the level of medical education they received from their infants’ provider [27].

Pediatric endocrinologists are involved in the initial assessment, but the long-term management of children with CH, which necessitates frequent follow-up and monitoring, may be conducted by pediatricians, nurse practitioners, or family physicians [28]. In a cross-sectional survey of primary care providers (PCPs) in California and Hawaii, the two most commonly perceived barriers to providing long-term care for patients with CH were (1) that they need guidance or support from endocrinologists and (2) that they were not familiar with the CH treatment guidelines. The proportion of PCPs who correctly identified the recommended frequency of blood tests for three different age groups was no higher than 73% [26]. In Alberta, infants who screen positive for CH receive an initial consultation with a pediatric endocrinologist for confirmation of the diagnosis and initiation of treatment. Pediatric endocrinologists are based in the two largest cities—Edmonton and Calgary. Subsequent follow-up may be performed by a pediatrician or family doctor. The administrative codes in the databases were not sufficiently granular to distinguish the specialty of the ordering provider.

Alberta has been screening infants for CH for almost 50 years, and it is fair to conclude from our study that most infants are benefiting from timely and appropriate long-term management. However, the handful of infants with scant monitoring or prolonged hypothyroidism spotlights gaps in care that call out for changes to how long-term follow-up happens. Future studies can follow patients prospectively through their clinical journeys. The ANSP can develop a program to track longitudinally the clinical outcomes, particularly of those with the most severe disease. The ANSP has also developed a Clinical Algorithm tool that can be more consistently distributed to providers and parents at the time of diagnosis [16]. Solutions will require a multi-pronged approach and include eliminating physical and figurative barriers (e.g., geographic distance and administrative red tape) that prevent infants from accessing healthcare providers and laboratory facilities; systematic, initial education and periodic re-education with primary care providers and caregivers; and facilitating access to long-term outcomes data for the purposes of quality improvement.

## 5. Conclusions

It is imperative that the newborn screening process for CH evaluate the outcomes of infants who have confirmed CH as part of continuous quality improvement. More work needs to be performed to understand why some are experiencing prolonged periods of hypothyroidism or infrequent lab monitoring. There should be a process for identifying and supporting the management of infants whose thyroid hormone levels are significantly abnormal.

## Figures and Tables

**Figure 1 IJNS-10-00035-f001:**
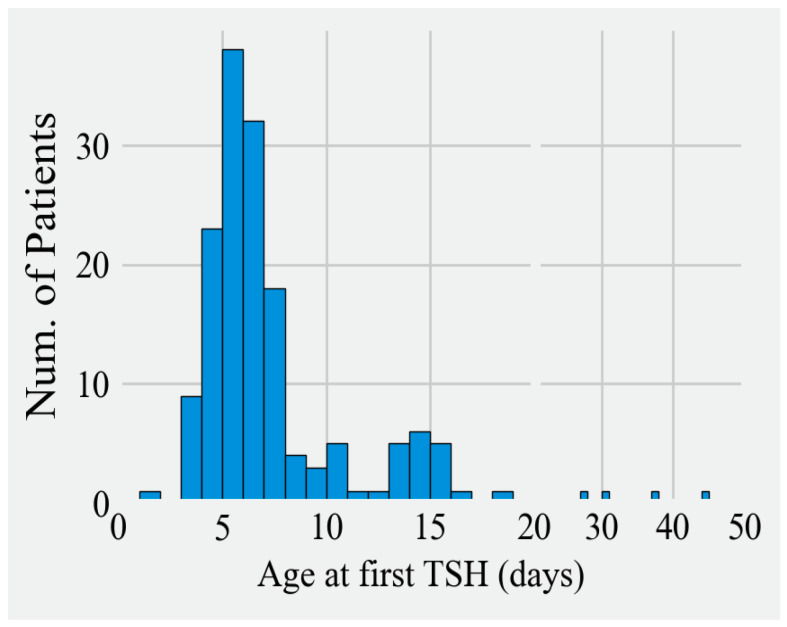
Age of infant in postnatal days at the time of first venous TSH measurement.

**Figure 2 IJNS-10-00035-f002:**
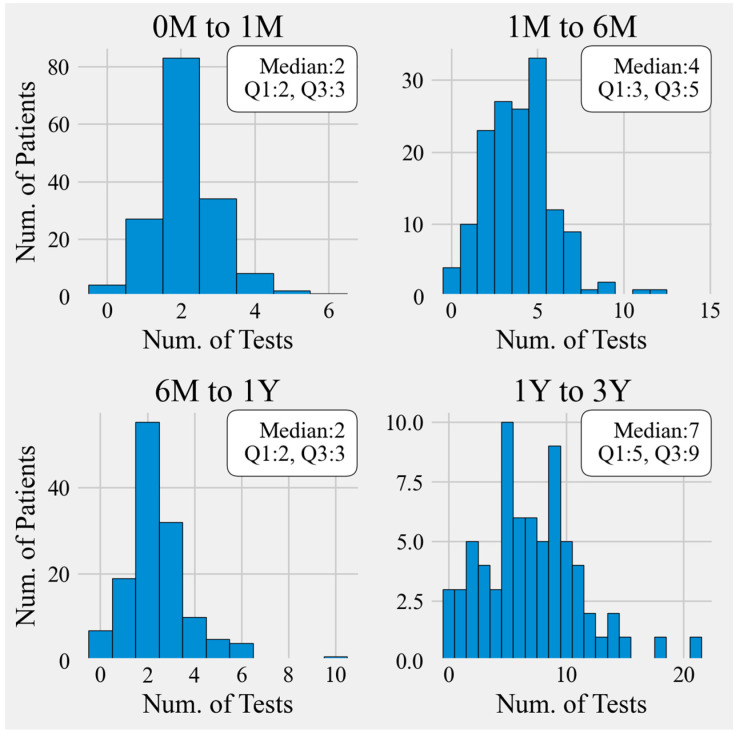
Number of TSH tests completed during specific age ranges. Abbreviations: TSH, Thyroid Stimulating Hormone; M, months; Y, years. Median (Q1, Q3) shows the 50th, 25th, and 75th-centiles of the range of results.

**Figure 3 IJNS-10-00035-f003:**
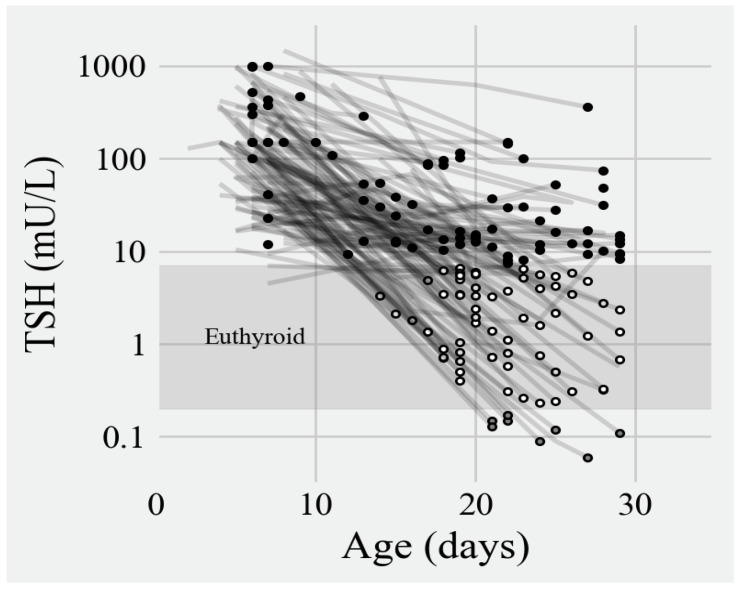
Range of initial and 1-month TSH levels. (Filled circles = hypothyroid; open circles = euthyroid; gray circles = hyperthyroid).

**Figure 4 IJNS-10-00035-f004:**
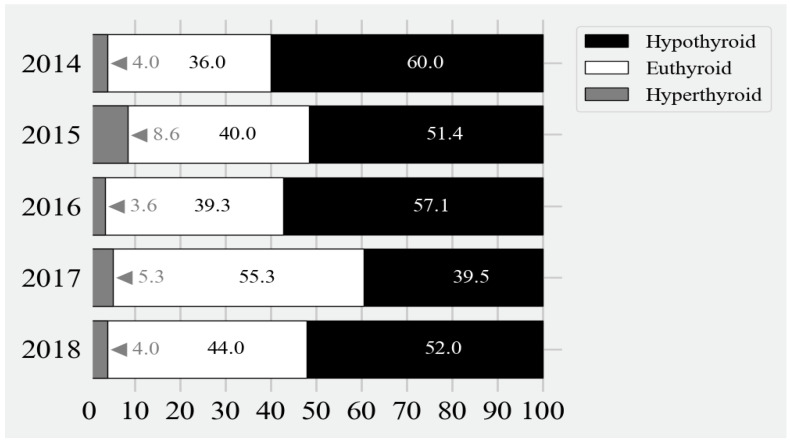
Frequencies (%) of euthyroidism, hypothyroidism, and hyperthyroidism at 1 month of age. Euthyroidism was defined as a TSH level between 0.2 and 7 mU/L, hypothyroidism defined as TSH > 7 mU/L, and hyperthyroidism defined as TSH < 0.2 mU/L. Year 2019 (infants born between 1 January and 31 March 2019) was not included in this analysis because there were insufficient data.

**Table 1 IJNS-10-00035-t001:** Age of infant in postnatal days at the time of first venous TSH measurement.

Year	2014 (*n* = 27)	2015 (*n* = 35)	2016 (*n* = 30)	2017 (*n* = 39)	2018 (*n* = 25)	2019 (*n* = 4)
Time to 1st TSH (Days)						
Range	5 to 16	2 to 28	4 to 45	4 to 38	4 to 28	7 to 11
Mode	6	8	6	5	6	N/A
Median (Q1, Q3)	7 (6, 8)	7 (6, 9)	6 (6, 7)	7 (5, 8)	6 (6, 8)	N/A

Median (Q1, Q3) shows the 50th, 25th, and 75th-centiles of the range of results. N/A: data not available for this time frame.

## Data Availability

Restrictions apply to the availability of these data. Data were obtained from Alberta Health and are available from the authors with the permission of Alberta Health.

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
