# Peer review of "Evaluation of the First Three Years of Treatment of Children with Congenital Hypothyroidism Identified through the Alberta Newborn Screening Program"

_2409-515X, 2024, doi:10.3390/ijns10020035_

Round 1
Reviewer 1 Report
Comments and Suggestions for Authors
The aim of this study was to evaluate the extent to which infants with congenital hypothyroidism (CH) have received timely and appropriate management within the first 3 years of life subsequent to their identification through the Alberta Newborn Screening Program for CH. To achieve this goal, the Authors analyzed data extracted from Alberta Health administrative databases for infants born between 2014 and 2019. They found that a high number of infants experienced prolonged periods of hypothyroidism or infrequent lab monitoring after receiving a confirmed diagnosis of CH. They therefore concluded that efforts are needed to improve the screening program for CH in Alberta.
Major observations
Materials and Methods
NBS for CH in Alberta. The NBS program for CH is too shortly described and more details are needed. For example, what is the TSH cutoff at the first screening and at re-screening, what are the TSH values corresponding to “borderline” and “critical” out-of-range results indicated at line 86, etc... It is not specified how many screening laboratories there are in Alberta and, if there are two or more labs, whether all of them follow the same screening procedures (same cutoffs? Same re-screening procedures?).
In addition, it is not clear whether all premature infants or only those with BW less than 2000 g are re-screened due to the risk of delayed TSH rise. At lines 89-92 it is written that all infants with BW less than 2000 g are recommended to have blood spots recollected at 21–28 days of age, but at lines 123-130 the Authors report that NBS in premature infants (with no limitation of BW) is repeated at 21-28 days of life if the initial screen was normal. This point needs to be clarified.
Results
Incidence. In the Introduction the Authors specify that the CH incidence in Alberta has been reported to be 1:3500 based on data collected in the period 2002-2005 (ref#5). Nevertheless, the results obtained in this study show an incidence (1:1470) significantly higher than that previously estimated and more similar to that recently reported in other countries. This finding is very important and should be emphasized in the discussion.
In addition, since the incidence estimate represented an important part of the analysis of data, it could be included among the objectives of the study.
Table 1. To have a more complete information on the distribution of days at the first venous TSH measurement, the row indicating the median value in each year should also include the values corresponding to the first and the third quartile. Data can be presented as “median; Q1, Q2”.
Figure 1. This figure appears redundant because it does not provide any further information with respect to Table 1. I suggest to remove it.
Timeliness of diagnosis. No data on the age at the start of the replacement therapy are provided. Since this variable is an important indicator of effectiveness of a screening program for CH, median age at start of treatment should be provided either for term or preterm infants diagnosed with CH. Moreover, no data on different etiology of CH are provided (agenesis, ectopy, hypoplastic, hyperplastic, normal thyroid). This information is important not only to calculate the prevalence of different diagnoses, but also to evaluate whether infants with thyroid dysgenesis (the more severe forms of CH) are treated precociously.
Figure 2. To make this figure more informative, in each panel the number of TSH tests corresponding to Q1 and Q3 could be shown along with median value.
Blood TSH and serum TSH. To avoid any confusion with blood TSH at screening, specify “serum TSH (sTSH)” through the text when appropriate (lines 110, 115, 119, 120, etc....).
Timeliness of monitoring. At lines 173-179 cases with initial high serum TSH values and infrequent laboratory monitoring are described. However, it is not specified whether these infants had never started the replacement therapy and at what age.
Discussion
At lines 233-234 it is written “TSH levels at 1 month of age were generally lower and/or within normal range, especially for those with the highest initial TSH levels, implying that levothyroxine treatment had been initiated.” This sentence needs to be modified because it is conflicting with data shown in Fig.3 and Fig.4, indicating that around half of the infants with confirmed CH were still hypothyroid (some severely hypothyroid) at 1 month of life.
The prolonged hypothyroid condition in infants diagnosed with CH may be essentially due to a delayed start of the replacement therapy or an inadequate dosage of levothyroxine. Is there a way to verify these hypotheses?
Similarly, at lines 252-253 and 277 it is reported that subclinical hypothyroidism at 1 month of life is common in the studied cohort. Nevertheless, it should be also added that around one third of the infants were still severely hypothyroid at that age, as clearly shown in Figure 3. Prolonged periods of severe or subclinical hypothyroidism represent a very critical issue of the screening program in Alberta. This point should be clearly reported and discussed to draw attention on a too high number of CH infants experiencing this condition.
Author Response
Dear Reviewer,
Thank you for your comments.
Comment #1: Materials and Methods
NBS for CH in Alberta.
The NBS program for CH is too shortly described and more details are needed. For example, what is theTSH cutoff at the first screening and at re-screening, what are theTSH values corresponding to “borderline” and “critical” out-of-range results indicated at line 86, etc... It is not specified how many screening laboratories there are in Alberta and, if there are two or more labs, whether all of them follow the same screening procedures (same cutoffs? Same re-screening procedures?).
In addition, it is not clear whether all premature infants or only those with BW less than 2000 g are re-screened due to the risk of delayed TSH rise. At lines 89-92 it is written that all infants with BW less than 2000 g are recommended to have blood spots recollected at 21–28 days of age, but at lines 123-130 the Authors report that NBS in premature infants (with no limitation of BW) is repeated at 21-28 days of life if the initial screen was normal. This point needs to be clarified.
Response #1: A more detailed description of NBS in Alberta, including TSH cutoffs has been added to the “Material and Methods”.
[lines 79 to 82 in revised manuscript] The ANSP is a province-wide screening program offered to all infants born in Alberta and is located at the University of Alberta in the provincial capital, Edmonton. The ANSP screens for 22 conditions, including Congenital Hypothyroidism due to primary causes. The NBS for CH is performed by measuring…
[lines 89 to 92 in revised manuscript] The CH screening algorithm has two types of out-of-range results: borderline or critical. The borderline cutoff is age-related (0-30 hours: TSH ≥32 mU/L; 31-191 hours: TSH ≥26 mU/L; ≥ 192 hours: TSH ≥20 mU/L). The critical cutoff (TSH ≥50 mU/L) is the same for all age groups.
[lines 94 to 95 in revised manuscript] Two borderline results (on separate specimens) are followed up as a critical result.
Response #1 (Cont):
[lines 131 to 132 in revised manuscript] In all premature infants with BW <2000g, their NBS are repeated between 21 and 28 days of age if the initial screen was normal.
Comment #2: Results
Incidence.
In the Introduction the Authors specify that the CH incidence in Alberta has been reported to be 1:3500 based on data collected in the period 2002-2005 (ref#5). Nevertheless, the results obtained in this study show an incidence (1:1470) significantly higher than that previously estimated and more similar to that recently reported in other countries. This finding is very important and should be emphasized in the discussion. In addition, since the incidence estimate represented an important part of the analysis of data, it could be included among the objectives of the study.
Response #2: Because determining this result was not a priori one of the study’s objectives, we initially did not emphasize it during the discussion. Based on Reviewer 1’s comment about the importance of this change in incidence, we agree that it is worthwhile to comment upon it in the discussion.
[lines 241 to 249 in revised manuscript] We noticed an increased incidence of CH in Alberta over the last 20 years. Between 2002-2005 the incidence of CH in Alberta had been reported to be approximately 1 in 3500; however, our study shows an incidence of 1:1470 [5]. An increased global incidence of CH over the past 2-3 decades has been reported also in the province of Quebec in Canada, as well as in France, New Zealand, Italy, Argentina, Greece, and the United States [8,9]. It is not completely clear what is causing this increase; most of these increases have been ascribed to lowering of TSH cut-offs, therefore allowing for detection of milder cases of CH. In some studies, changes in the ethnic composition of the screened population were also considered to be the cause of the increase [8,9].
Please note that two additional studies were cited and added to the references. Consequently, the references had to be re-numbered throughout the manuscript.
21. Barry, Y.; Bonaldi, C.; Goulet, V.; Coutant, R.; Leger, J.; Paty, A.C.; Delmas, D.; Cheillan, D.; Roussey, M. Increased incidence of congenital hypothyroidism in France from 1982 to 2012: a nationwide multicenter analysis. Ann Epidemiol 2016, 26, 100-105 e104, doi:10.1016/j.annepidem.2015.11.005.
22. Deladoey, J.; Ruel, J.; Giguere, Y.; Van Vliet, G. Is the incidence of congenital hypothyroidism really increasing? A 20-year retrospective population-based study in Quebec. J Clin Endocrinol Metab 2011, 96, 2422-2429, doi:10.1210/jc.2011-1073.
Comment #3: Table 1
This Reviewer has suggested supplementing the data about median values with the interquartile ranges in Table 1 and removing Figure 1.
Response #3: These data have been added to Table 1 and are shown as median (Q1, Q3) [lines 164 to 167 in revised manuscript]. (See attached file)
Comment #4: Figure 1
Reply #4: We retained Figure 1 in the revised manuscript, contrary to Reviewer 1’s suggestion, because it visually provides information about the distribution of ages at first serum TSH measurement across all years. We followed the Academic Editor’s advice, as well, to retain Figure 1.
Comment #5: No data on the age at the start of the replacement therapy are provided. No data on different etiology of CH are provided.
Response #5: The Reviewer has commented rightly that there are no data regarding when levothyroxine therapy was started, nor any information about the etiology of the primary congenital hypothyroidism. Because we relied on retrospective data from administrative databases, the data regarding timing of levothyroxine starts were incomplete. In addition, these administrative databases did not include information about the etiology of hypothyroidism, nor did the study aim to determine the prevalence of these various etiologies.
The points raised by the Reviewer are well-taken and are limitations of the study. Therefore, we have edited the portion of the discussion:
[lines 296 to 303 in revised manuscript] It can be inferred that the overall lowering of TSH levels after confirmation of diagnosis reflects that therapy was started; however, the timing of when levothyroxine therapy was prescribed and started could not be accurately ascertained from the administrative databases and is a limitation of this study. While inadequate therapy could be one reason for suboptimal outcomes, the reasons are likely to be multifactorial. These include the etiology of congenital hypothyroidism, with thyroid agenesis, ectopia, or hypoplasia likely to result in more severe disease; limited access to healthcare providers, laboratory facilities, and treatments; parental non-adherence due to a misunderstanding…
Comment #6: The Reviewer has requested that Figure 2 also display Q1 and Q3 in each panel.
Reply #6: These data have been added.
[lines 176 to 180 in revised manuscript] (see Attached File)
Figure 2. Number of TSH tests completed during specific age ranges. Abbreviations: (TSH, Thyroid Stimulating Hormone; M, months; Y, years). Median (Q1,Q3) shows the 50th,25th, and 75th-centiles of the range of results
Comment #7: Blood TSH and serum TSH.
Reply #7: Regarding the “Appropriateness of treatment” section, the Reviewer has suggested using the abbreviation sTSH to indicate serum TSH and avoid confusion with blood TSH collected at the time of screening. We respectfully did not make the change because only serum TSH is measured to monitor treatment. We added “serum” [to lines 109 and 110 in previous manuscript, now lines 115 and 116 in revised manuscript] to clarify that serum free T4 and serum TSH levels should be monitored.
The goals are to normalize serum FT4 levels within 2 weeks of age and serum TSH levels within 1 month of age.
Comment #8: It is not specified whether these infants had never started the replacement therapy and at what age.
Reply #8: Regarding lines 173-179 (previous manuscript version), we can only describe our findings and speculate (in the Discussion) regarding reasons for infrequent monitoring or suboptimal TSH levels. The administrative databases could not provide sufficiently granular data for all the infants to ascertain when treatment was started or at what dose. (Please see earlier comment). These cases underscore the importance of establishing a means of prospectively and longitudinally tracking these infants’ clinical courses and having a way to flag when TSH levels are suboptimal.
Comment #9: Discussion
At lines 233-234 it is written “TSH levels at 1 month of age were generally lower and/or within normal range, especially for those with the highest initial TSH levels, implying that levothyroxine treatment had been initiated.” This sentence needs to be modified because it is conflicting with data shown in Fig 3. and Fig 4. Indicating that around half of the infants with confirmed CH were still hypothyroid at 1 month of life.
Reply #9: With respect to the need for clarification of lines 233-234 (previous manuscript version) – “TSH levels at 1 month of age were generally lower and/or within normal range, especially for those with the highest initial TSH levels, implying that levothyroxine treatment had been initiated” – this sentence was an attempt to suggest that treatment was likely to have been started for the majority of infants for whom CH had been confirmed because there was a general lowering of the TSH levels. We have edited the sentence and hope that it is more clear to its purpose:
[lines 251 to 254 in revised manuscript] There was an overall decrease in TSH levels between the time of the initial confirmatory serum sample and one month of age. Though we cannot with certainty state that this was due to levothyroxine therapy, the trend implies that treatment had been started.
Reviewer 2 Report
Comments and Suggestions for Authors
The paper examines the management of congenital hypothyroidism (CH) in infants within the first three years of life following diagnosis through newborn screening (NBS) in Alberta, Canada. The study analyzes data from 160 infants with CH, revealing variations in the timeliness of diagnosis (95% in 16 days), monitoring frequency (median meet), and appropriateness of treatment (50%). While most infants received timely confirmation of diagnosis and appropriate treatment initiation, some experienced prolonged periods of hypothyroidism or inadequate monitoring.
The paper provides valuable insights into the management of CH following NBS and helps us understand the region's performance. However, the retrospective nature of the study, using administrative data, limits the depth of understanding regarding the reasons behind variations in monitoring frequency and treatment appropriateness.
In addition, it doesn’t contain data such as the final diagnosis and the outcomes, IQ, to answer whether this practice is appropriate for babies with hypothyroidism detected as newborns. The incidence is 1 in 1470, far from the literature on congenital hypothyroidism. It might contain both permeant and transient hypothyroidism.
Author Response
Dear Reviewers #2:
We would like to thank you for your positive and constructive comments on our manuscript. All comments and suggestions have been addressed in the revised manuscript (titled Manuscript version 2). Below is a detailed response to your comment and the changes made to the manuscript. Please note that we have highlighted in yellow any newly added or edited text in the revised manuscript.
With sincere gratitude,
Elizabeth T. Rosolowsky, MD, MPH,
Corresponding author
Comment #1: It doesn't contain data such as the final diagnosis and the outcomes, IQ, to answer whether this practice is appropriate for babies with hypothyroidism detected as newborns. The incidence is 1 in 1470, far from the literature on congenital hypothyroidism. It might contain both permanent and transient hypothyroidism.
Reply #1: Reviewer 2 has commented rightly that there are no data regarding the etiology of the primary congenital hypothyroidism. The administrative databases from which we extracted data did not include information about the etiologies of hypothyroidism, nor did the study aim to determine the prevalence of these various etiologies.
The points raised by Reviewer 2 are well-taken and are limitations of the study. Therefore, we have edited the portion of the discussion:
[lines 296 to 303 in revised manuscript] It can be inferred that the overall lowering of TSH levels after confirmation of diagnosis reflects that therapy was started; however, the timing of when levothyroxine therapy was prescribed and started could not be accurately ascertained from the administrative databases and is a limitation of this study. While inadequate therapy could be one reason for suboptimal outcomes, the reasons are likely to be multifactorial. These include the etiology of congenital hypothyroidism, with thyroid agenesis, ectopia, or hypoplasia likely to result in more severe disease; limited access to healthcare providers, laboratory facilities, and treatments; parental non-adherence due to a misunderstanding…
Reply #1 (continued): Because determining the incidence was not a priori one of the study’s objectives, we initially did not emphasize it during the discussion. Based on Reviewer 2's comments about the importance of this change in incidence, we agree that it is worthwhile to comment upon it in the discussion.
[lines 241 to 249 in revised manuscript] We noticed an increased incidence of CH in Alberta over the last 20 years. Between 2002-2005 the incidence of CH in Alberta had been reported to be approximately 1 in 3500; however, our study shows an incidence of 1:1470 [5]. An increased global incidence of CH over the past 2-3 decades has been reported also in the province of Quebec in Canada, as well as in France, New Zealand, Italy, Argentina, Greece, and the United States [8,9]. It is not completely clear what is causing this increase; most of these increases have been ascribed to lowering of TSH cut-offs, therefore allowing for detection of milder cases of CH. In some studies, changes in the ethnic composition of the screened population were also considered to be the cause of the increase [8,9].
Please note that two additional studies were cited and added to the references. Consequently, the references had to be re-numbered throughout the manuscript.
- Barry, Y.; Bonaldi, C.; Goulet, V.; Coutant, R.; Leger, J.; Paty, A.C.; Delmas, D.; Cheillan, D.; Roussey, M. Increased incidence of congenital hypothyroidism in France from 1982 to 2012: a nationwide multicenter analysis. Ann Epidemiol 2016, 26, 100-105 e104, doi:10.1016/j.annepidem.2015.11.005.
- Deladoey, J.; Ruel, J.; Giguere, Y.; Van Vliet, G. Is the incidence of congenital hypothyroidism really increasing? A 20-year retrospective population-based study in Quebec. J Clin Endocrinol Metab 2011, 96, 2422-2429, doi:10.1210/jc.2011-1073.
Reviewer 3 Report
Comments and Suggestions for Authors
The manuscript "Evaluation of the first three years of treatment of children with Congenital Hypothyroidism identified through the Alberta Newborn Screening Program" by Sosova et al. is an important contribution to the field of neonatal screening and endocrinology, specifically addressing congenital hypothyroidism (CH) management in the Alberta region. The study leverages a retrospective chart review methodology to evaluate the timeliness and appropriateness of diagnosis and treatment of CH within the first three years of life, utilizing deidentified laboratory data extracted from Alberta Health administrative databases. The study addresses crucial aspects of CH management—timeliness of diagnosis and ongoing monitoring—which are essential for preventing long-term neurodevelopmental consequences. The findings that 95% of infants had biochemical confirmation of diagnosis by 16 days of age and the detailed analysis of TSH monitoring intervals provide valuable insights into practice patterns and areas for improvement.
Minor suggestions:
- Please add the precise protocol of CH newborn screening in Alberta to better understand the usual requirements (e.g. what is borderline and what is critical TSH range; same cut-off levels for 24h and for 72h?)
- The study notes variability in the frequency of monitoring and attainment of euthyroidism but does not deeply analyze the underlying factors contributing to these variations or even to examine the (not so few) cases with significant high TSH levels during the studied period - any option to further explore circumstances in these cases (possibly in following research)?
- Although the study concludes with the need for a multi-pronged approach to address gaps in care, specific, actionable recommendations are somewhat lacking. Providing concrete suggestions for policy changes, educational programs for healthcare providers and parents, or the development of a centralized monitoring system could offer a clearer path forward for improving CH management.
Author Response
Dear Reviewer #3:
We would like to thank you for your positive and constructive comments on our manuscript. All comments and suggestions have been addressed in the revised manuscript (titled Manuscript version 2). Below is a detailed response to your comment and the changes made to the manuscript. Please note that we have highlighted in yellow any newly added or edited text in the revised manuscript.
With sincere gratitude,
Elizabeth T. Rosolowsky, MD, MPH,
Corresponding author
Comment #1: Please add the precise protocol of CH newborn screening in Alberta to better understand the usual requirements (e.g. what is borderline and what is critical TSH range; same cut-off levels for 24h and for 72h?)
A more detailed description of NBS in Alberta, including TSH cutoffs has been added to the “Material and Methods”.
[lines 79 to 82 in revised manuscript] The ANSP is a province-wide screening program offered to all infants born in Alberta and is located at the University of Alberta in the provincial capital, Edmonton. The ANSP screens for 22 conditions, including Congenital Hypothyroidism due to primary causes. The NBS for CH is performed by measuring…
[lines 89 to 92 in revised manuscript] The CH screening algorithm has two types of out-of-range results: borderline or critical. The borderline cutoff is age-related (0-30 hours: TSH ≥32 mU/L; 31-191 hours: TSH ≥26 mU/L; ≥ 192 hours: TSH ≥20 mU/L). The critical cutoff (TSH ≥50 mU/L) is the same for all age groups.
[lines 94 to 95 in revised manuscript] Two borderline results (on separate specimens) are followed up as a critical result.
Comment #2: The study notes variability in the frequency of monitoring and attainment of euthyroidism but does not deeply analyze the underlying factors contributing to these variations or even to examine the (not so few) cases with significant high TSH levels during the studied period – any option to further explore circumstances in these cases (possibly in following research)?
Reply #2: The results of this study have prompted us to go into further depth to try to understand the factors that may have contributed to suboptimal outcomes.
Comment #3: Although the study concludes with the need for a multi-pronged approach to address gaps in care, specific, actionable recommendations are somewhat lacking. Providing concrete suggestions for policy changes, educational programs for healthcare providers and parents, or the development of a centralized monitoring system could offer a clearer path forward for improving CH management.
Reply #3: We have made some concrete suggestions and added them to the discussion section:
[lines 334 to 338 in revised manuscript] Future studies can follow patients prospectively through their clinical journey. The ANSP can develop a program to track longitudinally the clinical outcomes, particularly of those with the most severe disease. The ANSP has also developed a Clinical Algorithm tool that can be more consistently distributed to providers and parents at the time of diagnosis [16].